# Predictive Factors of Complete Response to Transarterial Chemoembolization in Intermediate Stage Hepatocellular Carcinoma beyond Up-To-7 Criteria

**DOI:** 10.3390/cancers15092609

**Published:** 2023-05-04

**Authors:** Natsuhiko Saito, Hideyuki Nishiofuku, Takeshi Sato, Shinsaku Maeda, Kiyoyuki Minamiguchi, Ryosuke Taiji, Takeshi Matsumoto, Yuto Chanoki, Tetsuya Tachiiri, Hideki Kunichika, Nagaaki Marugami, Toshihiro Tanaka

**Affiliations:** 1Department of Diagnostic and Interventional Radiology, Nara Medical University, Shijyocho 840, Kashihara City 634-8522, Japan; summernatsu@naramed-u.ac.jp (N.S.); hmn@naramed-u.ac.jp (H.N.); satotake@naramed-u.ac.jp (T.S.); shinsaku@naramed-u.ac.jp (S.M.); kiyo829@naramed-u.ac.jp (K.M.); rtaiji@naramed-u.ac.jp (R.T.); t.matsumoto@naramed-u.ac.jp (T.M.); y.chanoki@naramed-u.ac.jp (Y.C.); k135334@naramed-u.ac.jp (T.T.); k102972@naramed-u.ac.jp (H.K.); marugami@naramed-u.ac.jp (N.M.); 2Department of Radiology, Higashiosaka City Medical Center, Nishiiwata 3-4-5, Higashiosaka City 578-8588, Japan

**Keywords:** hepatocellular carcinoma, intermediate stage, up-to-7 criteria, transarterial chemoembolization

## Abstract

**Simple Summary:**

To date, the indication for transarterial chemoembolization (TACE) of intermediate stage hepatocellular carcinoma (HCC) with a high tumor burden remains controversial. TACE has the advantage of a high possibility of achieving complete response (CR). This study aimed to clarify prognosis and identify predictors of CR by TACE in intermediate stage HCC beyond up-to-7 criteria. Overall, 56.9% of patients obtained CR, which contributed to prolonging survival. On multivariate analysis, the predictor of CR was HCC within up-to-11 criteria. In this study, the treatment strategy for intermediate stage HCC based on the 7–11 criteria is suggested.

**Abstract:**

Aim: To clarify the prognosis and identify predictors for obtaining a complete response (CR) by transarterial chemoembolization (TACE) in intermediate stage HCC beyond up-to-7 criteria. Methods: Of the 120 patients with intermediate stage HCC who were treated by TACE as the initial treatment from February 2007 to January 2016, 72 finally matched the following inclusion criteria: beyond up-to-7 criteria; Child–Pugh score under 7; and no combined therapy within 4 weeks after the initial TACE. The CR rate and overall survival (OS) were evaluated. Logistic regression analysis was performed to identify predictors of CR. The deterioration of liver function after TACE was also evaluated. Results: The CR rate was 56.9%, and the overall median survival time (MST) was 37.7 months. The MST was 38.7 months in the CR group and 28.0 months in the non-CR group (*p* = 0.018). HCC within up-to-11 criteria was the only predictor of CR. The CR rate and MST were 70.7% and 37.7 months, respectively, in patients with HCC within up-to-11 criteria and 38.7% and 32.7 months, respectively, in the patients beyond up-to-11 criteria. Deterioration of the Child–Pugh score after the initial TACE and the 2nd TACE occurred in 24.2% and 12.0%, respectively, and deterioration of the modified albumin-bilirubin (mALBI) grade occurred in 17.6% and 7.4%, respectively. Conclusion: TACE can achieve high CR rates with prolonged overall survival for intermediate stage HCC beyond up-to-7 criteria. The predictor of CR was within up-to-11 criteria. Deterioration of liver function was not severe, but requires caution. Multidisciplinary approach as additional treatment after TACE is important.

## 1. Introduction

Transarterial chemoembolization (TACE) has been the standard therapeutic option for intermediate stage hepatocellular carcinoma (HCC) of the Barcelona Clinic Liver Cancer (BCLC) staging system [1]. Intermediate stage HCC is very heterogeneous in terms of tumor diameter, tumor counts, and liver function, and TACE is not beneficial for all patients. Due to the recent development of molecular targeted agents (MTAs) and immunotherapies, the role of TACE has been discussed. The Asia-Pacific Primary Liver Cancer Expert Meeting (APPLE) 2019 proposed that HCC “beyond up-to-7 criteria” was “likely to develop TACE failure/refractoriness” [2]. Other reports recommended “up-to-11 criteria” or “7–11 criteria” rather than up-to-7 criteria [3,4]. Thus, to date, the appropriate indication for TACE remains controversial.

On the other hand, expectations are currently rising for combined therapies of TACE with modern systemic therapies. Recent published literature demonstrated that combination therapies were more effective than TACE alone or systemic therapies [5,6,7,8]. In particular, for patients with high tumor burden, HCC––i.e., that beyond up-to-7 criteria––combination therapy could be beneficial. Therefore, it is important to determine which patients will benefit from TACE with systemic therapies.

The most attractive advantage of TACE is the high possibility of achieving complete response (CR). Patients can have a chance of a treatment-free period after achieving CR, which could prolong overall survival. In 2015, Kim et al. reported that patients who obtained CR in the initial TACE had a long survival, including a large number of patients with early stage HCC [9]. In 2020, Park et al. reported that CR as the initial response and best response during repeated TACE prolonged overall survival equally in patients with intermediate stage HCC [10]. To date, whether TACE could achieve CR in HCC beyond the up-to-7 criteria remains questionable. In addition, it is uncertain, in such high tumor burden cases, if the achievement of CR could prolong overall survival (OS). In addition, whether frequent TACE would cause deterioration of liver function in these high tumor burden cases is unknown, although it was previously shown to be safe for intermediate stage HCC [11].

From the above background, the purpose of this study was to investigate the CR rate by TACE in intermediate stage HCC beyond the up-to-7 criteria and to evaluate its prognosis. Further, the predictors of CR in these patients were investigated. Finally, the effect of TACE on liver function was evaluated.

## 2. Materials and Methods

### 2.1. Patient Data Collection and Eligibility Criteria

Our institutional review board approved this single-arm retrospective study. This study was carried out ethically in accordance with the Declaration of Helsinki of the World Medical Association. Informed consent was obtained from each patient before the TACE procedure.

The database in our institution was retrospectively reviewed. The inclusion criteria were as follows: (a) diagnosis of HCC established on the basis of findings from computed tomography (CT) and/or magnetic resonance imaging (MRI); (b) staging of HCC was classified as intermediate stage (BCLC-B, a solitary tumor over 5 cm in diameter was not included); (c) the initial TACE was performed during February 2007 to January 2016; (d) no prior treatment before the initial TACE; and (e) the sum of tumor counts and diameter was over 7 (beyond up-to-7 criteria).

The exclusion criteria were as follows: (a) Child–Pugh score over 8; (b) contrast-enhanced CT or MRI not obtained 1–3 months after TACE; and (c) other therapies for HCC, i.e., radiofrequency ablation (RFA), MTAs, surgical resection, or radiation therapy, given within 4 weeks after the initial TACE.

### 2.2. TACE Procedure

A combined CT and angiography system (Angio-CT System, Infinix Activ; Canon Medical Systems, Ohtawara, Japan) was used for all TACE procedures. CT during hepatic arteriography (CTHA) and CT during arterial portography (CTAP) were obtained. CTHA-maximum intensity projection (MIP) images were created using a 3D-CT workstation (Ziostation; Ziosoft, Tokyo, Japan in 2007 and Synapse Vincent; Fujifilm, Tokyo, Japan after 2008 until 2016) for TACE navigation. A 1.5 to 2.0-Fr tip microcatheter was inserted into the tumor feeding branches as selectively as possible.

### 2.3. Follow-Up and Evaluations

The tumor response was evaluated on contrast-enhanced CT or MRI obtained 1–3 months after TACE according to the modified Response Evaluation Criteria in Solid Tumors (mRECIST). Basically, in patients who had residual tumors after the initial TACE, a 2nd TACE was performed within a few weeks. For patients who achieved CR, CT or MRI was performed every 3 months. The Child–Pugh score and modified albumin-bilirubin (mALBI) grade were also evaluated every 3 months.

### 2.4. Definition

The CR group was defined as patients who obtained CR as the best response within 6 months after the initial TACE. For large HCC cases treated with bland TAE followed by conventional TACE (cTACE), the combination treatment was counted as one TACE procedure, and the treatment date was defined as that of the bland TAE. OS was defined as the time from the date of the initial TACE to the date of death. The clinical characteristics including age, sex, virus infection, Child–Pugh score, mALBI grade, α-fetoprotein (AFP), tumor diameter, tumor count, within up-to-11 criteria, and the use of MTAs were investigated.

### 2.5. Statistical Analysis

OS was calculated using the Kaplan–Meier method. The predictors of CR were investigated by univariate analysis using the χ2 test and multivariate analysis using logistic regression analysis. Factors with a *p* value < 0.1 on univariate analysis were included in the multivariate analysis. On multivariate analysis, *p* values of <0.05 were considered statistically significant. MST in two different groups was compared by the log-rank (Mantei-Cox) test. Statistical analyses were performed using SPSS, version 26.0 statistical software (SPSS Inc.; Chicago, IL, USA).

## 3. Results

### 3.1. Patients

There were 120 patients with beyond up-to-7 criteria intermediate stage HCC who received TACE as an initial treatment from February 2007 to January 2016, and 48 patients were excluded for the following reasons: 18 patients had a Child–Pugh score ≥ 8; 3 patients did not undergo contrast-enhanced CT or MRI after TACE; and 27 patients received other treatment, i.e., RFA, surgical resection, or radiation therapy, within 4 weeks after the initial TACE. Finally, 72 patients (59 men, 13 women; median age 75 years; range 50–89 years; 14 hepatitis B-virus positive; 33 hepatitis C virus-positive; 26 no-virus infection) were included in this study (Figure 1). A total of 65 patients (90.3%) were Child–Pugh score 5/6, and 7 patients (9.7%) were score 7. A total of 49 patients (68.1%) were mALBI grade 1/2a, and 23 patients (31.9%) were grade 2b. 59 patients (81.9%) were under the AFP level 200 ng/mL, and 13 patients (18.1%) were over AFP level 200 ng/mL. 41 patients (56.9%) were within up-to-11 criteria, and 31 patients (43.1%) were beyond up-to-11 criteria. The mean largest tumor diameter was 5.25 ± 2.60 cm (range: 1.5–14.0 cm). The mean tumor number was 5.71 ± 3.37 (range: 2–12 counts). The median follow-up period was 28.1 months (range 1.2–103.1 months). Of the 72 patients, 16 (22.2%) received an MTA, sorafenib, after being refractory to TACE. The patients’ demographic data are summarized in Table 1. In the initial TACE, cTACE was performed in 66 cases (epirubicin 63 and cisplatin 3), epirubicin-loaded drug-eluting beads (DC Beads; Boston Scientific, Marlborough, MA, USA) TACE (DEB-TACE) was performed in 2, cisplatin mixed with gelatin sponge particles (Gelpart; Nippon Kayaku) was performed in 3, and only bland-TAE using microspheres was performed in 1. Bland TAE prior to cTACE was performed in five cases of large size HCC. The median size was 12 cm (with a range of 7.3 cm to 14 cm). The embolization material comprised Embosphere (Merit Medical Japan, Tokyo, Japan) in two cases and gelatin sponge particles in the remaining three cases.

### 3.2. Tumor Response

CR was achieved in 35 patients in the initial TACE (48.6%), with partial response (PR) and stable disease (SD) in 30 patients (41.7%) and 7 patients (9.7%), respectively. A total of 27 patients underwent 2nd TACE within 6 months after the initial TACE; 23 had residual tumors, and 6 of them (26.1%) achieved CR in the 2nd TACE; and the remaining 4 had early recurrence after obtaining CR in the initial TACE, with all 4 (100%) obtained CR again in the 2nd TACE. Consequently, a total of 41 patients (56.9%) achieved the best response of CR within 6 months.

### 3.3. Rate of Liver Function Deterioration

Deterioration of liver function was calculated every time as follows: “number of patients deteriorated liver function 1–3 month after TACE/number of patients who received TACE” compared pre-TACE and post-TACE, Child–Pugh score increase, or mALBI grade decrease, which was defined as the deterioration of liver function. Deterioration of Child–Pugh score after the initial TACE and 2nd TACE was 24.2% and 12.0%, respectively. That of the mALBI grade was 17.6% and 7.4%, respectively.

### 3.4. OS Rates

The MST was 37.7 months. Survival rates at 1, 2, and 3 years were 89.4%, 69.3%, and 52.5%, respectively (Figure 2). The CR group demonstrated significantly longer OS compared with the non-CR group (38.7 vs. 28.0 months, *p* = 0.018) (Figure 3).

OS, overall survival; TACE, transarterial chemoembolization; HCC, hepatocellular carcinoma; CR, complete response; MST, median survival time.

### 3.5. Univariate and Multivariate Analyses

Results of univariate analysis showed that mALBI grade 1, 2a (*p* = 0.097), AFP < 200 ng/mL (*p* = 0.043) and within up-to-11 criteria (*p* = 0.008) were factors associated with inclusion in the CR group. On multivariate analysis, only within up-to-11 criteria (*p* = 0.042) remained a significant predictor of CR (Table 2). In the subgroup within up-to-11 criteria, the CR rate and OS were 70.7% and 37.7 months, respectively, whereas in the subgroup beyond up-to-11 criteria, they were 38.7% and 29.9 months, respectively.

### 3.6. Analyses of Additional Treatments after TACE

Out of 72 patients, 39 (54.2%) received additional other treatments after TACE. Sixteen patients received sorafenib, 13 received RFA, 9 received arterial infusion chemotherapy, and 2 received hepatic resection. The MST of the additional treatment group was 38.7 months, while the MST of the non-additional treatment group was 36.5 months (*p* = 0.07). Although not statistically significant, the additional treatment group showed a tendency towards longer survival.

## 4. Discussion

There are some reports about TACE for intermediate stage HCC beyond up-to-7 criteria. In a proof-of-concept study, Kudo et al. reported that the MST and ORR of TACE were 21.3 months and 33.3% respectively [12]. Meanwhile, Hung reported 22.3 months and 12% [4]. Other authors reported MSTs of 21.5, 19, and 34 months [3,13,14]. Our study revealed the MST and ORR of 37.7 months and 56.9%, respectively, which is superior to past reports. Recently, combining MTA with TACE become popular for the treatment of HCC beyond up-to-7 criteria. Amioka et.al reported that the MST of Lenvatinib plus TACE combination therapy achieved 45 months, which is superior to our results [15].

Previously, Kim et al., were the first to show that achievement of CR by TACE prolonged OS in HCC, including a large number of early stage cases [9]. Later, Park et al. reported that CR was obtained in 70.2% of the patients with intermediate stage HCC, and the achievement of CR was the significant prognostic factor for OS [10]. The significance of CR in patients with high tumor burden has so far been unknown. The present study showed that CR was the contributing factor to prolonging survival even in patients with HCC beyond up-to-7 criteria.

We considered that achieving CR with only one session of TACE could be difficult in patients with a high tumor burden. In addition, the follow-up interval varied from 1 to 3 months after TACE, which makes it necessary to perform at least two sessions of TACE within 6 months to assess the best response. Based on these factors, we determined CR as the best response achieved within 6 months after the initial TACE in this study.

Kudo et al. reported that lenvatinib achieved MST of 37.9 months in intermediate stage HCC beyond up-to-7 criteria [12]. It was also reported that atezolizumab plus bevacizumab achieved MST of 25.8 months in intermediate stage HCC [16]. In the present study, TACE achieved MST of 38.7 months in the CR group. Therefore, patients who achieved CR by TACE can be sequentially treated with lenvatinib or atezolizumab plus bevacizumab as maintenance therapy, which could further prolong their survival, and then, later, there could also be the opportunity to stop the systemic therapy and obtain a drug-free period. Recently, the TACTICS-L study showed that TACE combined with lenvatinib achieved long progression-free survival (PFS) and OS [17].

On the other hand, in patients who did not achieve CR, the MST was significantly shorter at 28 months. For such non-CR patients, intensive systemic therapies are required. In the present study, the best response of CR was defined as achievement of CR within 6 months from the initial TACE. This means that we consider that TACE should not be repeated if the tumors are uncontrolled and a switch to systemic therapies should occur earlier.

It is important to investigate predictors of CR. In the present study, within up-to-11 criteria was extracted as the predictor of CR. The group of within up-to-11 criteria showed a high CR rate of 70.7% and longer survival of 37.7 months. These results support the up-to-11 criteria reported from Korea [3] and the 7–11 criteria reported from Taiwan and China [4,13]. Patients with HCC beyond up-to-7 criteria and within up-to-11 criteria (between 7 to 11) have a high chance of CR, which could prolong their survival. The findings from the study suggest that up-to-11 criteria is the only contributing factor that leads to CR in HCC. By following up-to-11 criteria, we can increase the chance of achieving CR in patients with HCC, thereby improving their overall prognosis.

Previous reports demonstrated that tumor count, but not tumor size, was related to prognosis. Tumor number ≥ 11 was reported as a poor prognostic factor, although tumor size ≥ 6 cm was not a significant factor [11]. Therefore, in cases of beyond up-to-11 criteria, closer attention to tumor number than tumor size is needed.

There are other factors that affect the prognosis of patients with HCC treated by TACE. Previously published literature showed that heterogeneous enhancement in the early phase of contrast-enhanced CT was related to a poor prognosis [11,18]. Recently, it was reported that heterogenous intensity of tumors in the hepatobiliary phase on gadoxetic acid disodium-enhanced-MRI (EOB-MRI) was significantly associated with a poor prognosis after TACE [19]. On the other hand, hyperintensity in the hepatobiliary phase on EOB-MRI reflecting the activation of WNT/B-catenin was associated with shorter PFS and rapid tumor growth speed in the case of anti-PD-1/PD-L1 monotherapy [20]. These factors should be further investigated to determine the best therapeutic option for HCC with high tumor burden.

Deterioration of liver function caused by repeated TACE has been reported [21,22]. In the APPLE consensus statement, nodules beyond the up-to-7 criteria, and especially bilobar multifocal HCCs, are classed as likely to become CP-B or C after TACE [2]. In the present study, the Child–Pugh score and ALBI grade deteriorated in 24.2% and 17.6%, respectively, of patients after the initial TACE procedure. These rates were higher than in the previous report that evaluated patients in the overall intermediate stage [11]. Greater care regarding the deterioration of liver function should be taken in high tumor burden cases due to the large embolization area, and it should be minimized by using current TACE techniques, i.e., super-selective catheterization using a tiny tip microcatheter and under TACE navigation images created by a 3D-CT workstation [23,24].

Although TACE is the main treatment in intermediate stage HCC, a multidisciplinary approach is often necessary. In fact, according to our study, 54.2% of the patients received additional treatments after TACE. Surgery or RFA was performed for patients who achieved downstaging by TACE. Arterial infusion therapy or sorafenib was used for disease exacerbation. With the availability of new molecular targeted agents and immune check point inhibitors, there is potential for prolonged survival.

This study had some limitations. First, the results were obtained from a single-institution, retrospective, single-arm, and small number of analyses. Second, tumor counts or tumor diameter were not individually investigated. Third, morphological and pathological factors were not evaluated. Therefore, further investigation including a larger number of patients will be needed.

## 5. Conclusions

In conclusion, over 50% of patients with intermediate stage HCC beyond up-to-7 criteria obtained CR by TACE, which helped prolong survival. The predictor of CR was within up-to-11 criteria. Around 70% of patients with HCC within up-to-11 criteria achieved CR. The deterioration of liver function caused by repeated TACE was mild, but caution is required. From these results, we can consider a treatment strategy based on 7–11 criteria to choose or combine TACE with systemic therapies in intermediate stage HCC. A multidisciplinary approach for additional treatments after TACE is also important.

## Figures and Tables

**Figure 1 cancers-15-02609-f001:**
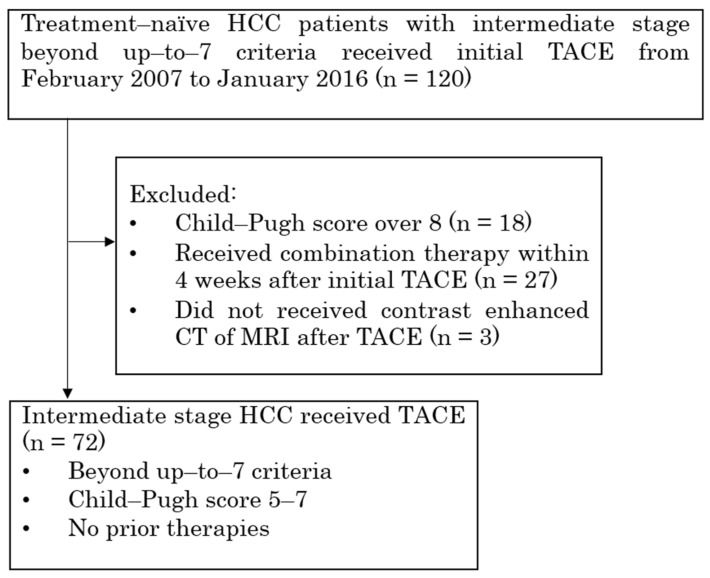
Flow diagram of study cohort who underwent TACE for HCC. TACE, transarterial chemoembolization; HCC, hepatocellular carcinoma.

**Figure 2 cancers-15-02609-f002:**
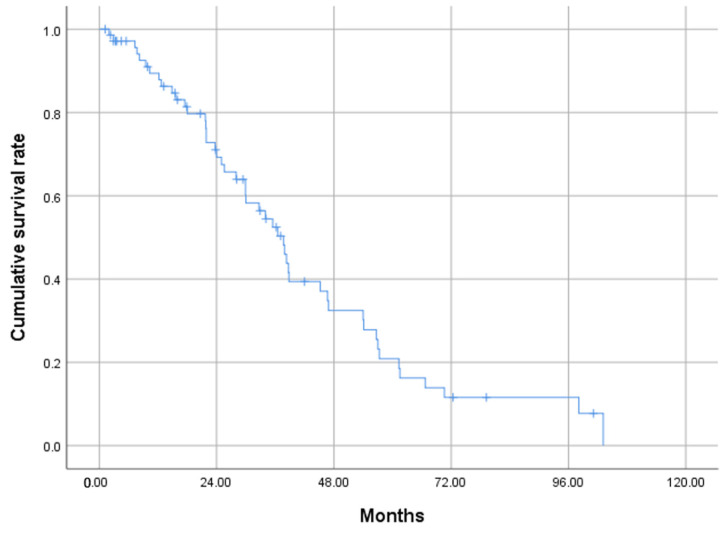
Cumulative OS of study patients who underwent TACE for HCC beyond the up-to-7 criteria, calculated by the Kaplan–Meier method. The MST is 37.7 months. The OS rates at 1, 2, and 3 years are 89.4%, 69.3%, and 52.5% respectively. OS, overall survival; TACE, transarterial chemoembolization; HCC, hepatocellular carcinoma; MST, median survival time.

**Figure 3 cancers-15-02609-f003:**
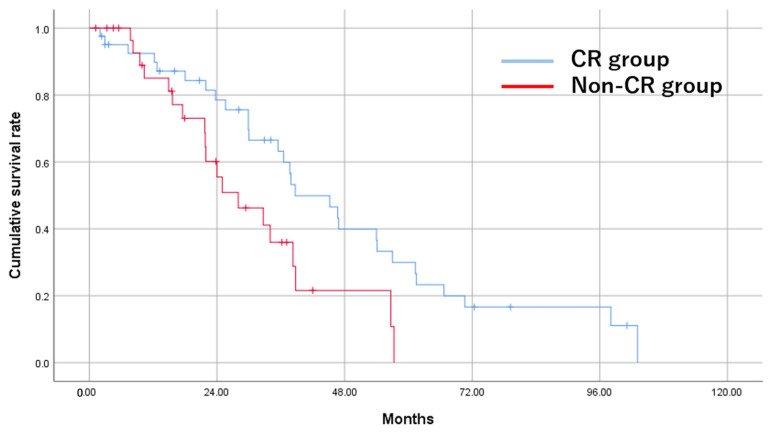
Cumulative OS of study patients who underwent TACE for HCC in the CR group and the non-CR group by the Kaplan–Meier method. The MST is 38.7 months for the CR group and 28.0 months for the non-CR group (*p* = 0.018).

**Table 1 cancers-15-02609-t001:** Summary of clinical characteristics of all patients.

Characteristics	Value	%
Age		
<75 years	39	54.2
≥75 years	33	45.8
Gender		
Male	59	81.9
Female	13	18.1
Virus		
B, C	46	63.9
nonB nonC	26	36.1
Child-Pugh score		
5, 6	65	90.3
7	7	9.7
mALBI grade		
1, 2a	49	68.1
2b	23	31.9
AFP (ng/mL)		
<200	59	81.9
≥200	13	18.1
Tumor diameter(cm)		
<6	51	70.8
≥6	21	29.2
Tumor count		
<7	47	65.2
≥7	25	34.8
Up-to-11 criteria		
in	41	56.9
out	31	43.1
MTA after refractory to TACE		
No	56	77.8
Yes	16	22.2

mALBI, modified albumin bilirubin; AFP, alpha-fetoprotein; MTA, molecular targeted agents.

**Table 2 cancers-15-02609-t002:** Univariate and Multivariate analysis for CR.

	Univariate Analysis		Multivariate Analysis	
Risk Factors	*p*	HR (95% CI)	*p*	HR (95% CI)
Age < 75 years	0.564	0.758 (0.296–1.942)		
Virus infection	0.737	0.844 (0.315–2.265)		
Child-Pugh score A	0.423	0.497 (0.090–2.749)		
mALBI grade 1, 2a	0.097	0.412 (0.145–1.172)	0.198	0.481 (0.158–1.466)
AFP < 200 ng/ml	0.043	3.784 (1.041–13.755)	0.121	3.022 (0.748–12.211)
Within up-to-11 criteria	0.008	0.261 (0.097–0.701)	0.042	0.341 (0.121–0.960)

mALBI, modified albumin bilirubin; AFP, alpha-fetoprotein.

## Data Availability

All data generated or analyzed during this study are included in this article.

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
