# Peer review of "Predictive Factors of Complete Response to Transarterial Chemoembolization in Intermediate Stage Hepatocellular Carcinoma beyond Up-To-7 Criteria"

_cancers, 2023, doi:10.3390/cancers15092609_

Round 1

Reviewer 1 Report

This article entitled “Clinical impact of complete response to transarterial chemoem- bolization in intermediate stage hepatocellular carcinoma be-3 yond the up-to-7 criteria” was aimed to evaluate the prognosis and identify predictors for obtaining a complete response (CR) by transarterial chemoembolization (TACE) in intermediate stage HCC beyond up-to-7 criteria.

My comments were listed as below.

1. In page 3, line 114,“The CR group was defined as patients who obtained CR as the best response within 6 months after the initial TACE.” Why did authors definite patient who achieved CR within 6 months after the initial TACE instead of patients who achieved CR immediately after first TACE or achieved CR after at least two sessions of TACE?  

2.The data of table1 did not show the information provided in Results 3.1 and 3.2.

3.The prognosis and data of these 72 patients did not be stratified by their additional treatments, such as RFA, arterial infusion therapy, and sorafenib. Therefore, authors please clarify the impact of these confounding factors on survival of patients enrolled in this study. Thus, authors please give more details of Table 2.

4.Thus, this study did not provide the solid evidence to support authors’ conclusion ( abstract, line 37 to 40).

Author Response

Reviewer 1

1. In page 3, line 114,“The CR group was defined as patients who obtained CR as the best response within 6 months after the initial TACE.” Why did authors definite patient who achieved CR within 6 months after the initial TACE instead of patients who achieved CR immediately after first TACE or achieved CR after at least two sessions of TACE?

Response>

Thank you for your attention. We added a paragraph explaining the reason why the definition of CR was achieved “2 sessions within 6 months” in discussion session.

2.The data of table1 did not show the information provided in Results 3.1 and 3.2.

Response>

Thank you for your attention. We reflected the data of table1 to Results 3.1.We added below information to Results 3.1.

65 patients (90.3%) are Child-Pugh score 5/6, and 7 patients (9.7%) are score 7. 49 patients (68.1%) are mALBI score 1/2a, and 23 patients (31.9%) are score 2b. 59 patients (81.9%) are under the AFP level 200 ng/ml, and 13 patients (18.1%) are over AFP level 200 ng/ml. 41 patients (56.9%) are within up-to-11 criteria, and 31 patients (43.1%) are beyond up-to-11 criteria.

3.The prognosis and data of these 72 patients did not be stratified by their additional treatments, such as RFA, arterial infusion therapy, and sorafenib. Therefore, authors please clarify the impact of these confounding factors on survival of patients enrolled in this study. Thus, authors please give more details of Table 2.

4.Thus, this study did not provide the solid evidence to support authors’ conclusion ( abstract, line 37 to 40).

Response>

The multivariate analysis of this study is to determine what kind of factor contribute to not “survival” but “CR by TACE”. Therefore, the results from multivariate analysis does not reflect the survival.

In addition, we tried to calculate the effectiveness of post TACE treatment such as sorafenib, HAIC, RFA and surgery for survival. The result was that MST of post TACE treatment group was 38.7 months versus non-post TACE treatment group was 36.5 months, which was not statistically significant (P=0.07). However, the MST was slightly longer in post TACE treatment group. Therefore, we consider that multidisciplinary approach for intermediate stage HCC beyond up-to-7 criteria treated after TACE is important. We added these results in result paragraph and the importance of multidisciplinary approach in abstract, discussion and conclusion.

Reviewer 2 Report

In this study, Saito and collaborators explored the TACE treatment strategy for intermediate-stage HCC based on the up-to-7 criteria. Therefore, of 120 patients treated by TACE as initial treatment, 72 matched with inclusion criteria of the study. The analysis showed that the CR rate was 56.9%, and the MST was 37.7 months compared to 28 months in the non-CR group. However, some key points that need further discussion are listed below:

Major comments

-Several reports have discussed the potential benefits of TACE treatment for intermediate HCC based on up to 7 criteria (PMID: 29278654, 34950185, 29278654, 31932243, etc.). Hence, the originality of this manuscript needs to be clarified, and the differences between this study and other cohorts must be described and discussed to explain the relevant contribution to the field.

-Previous reports have analyzed TACE treatment for intermediate HCC using different MTAs and immunotherapies. In this study, most cases of the CR group were treated with epirubicin (63) and cisplatin (3); however, there is not enough discussion of these results compared with studies that have used these drugs or others in TACE treatment for intermediate HCC. Therefore, the originality and relevance of the study must be clarified.

-The study design has severe limitations, including the number of patients, single institution, etc. Therefore, require an exhaustive evaluation since the results only suggest its potential benefit in the studied cohort but not a clinical impact that it contributes to the treatment of intermediate HCC.

Minor comments

In line 157, I suggest describing the significate of PR (partial response) and SD (stable disease) abbreviations.

Author Response

Major comments

-Several reports have discussed the potential benefits of TACE treatment for intermediate HCC based on up to 7 criteria (PMID: 29278654, 34950185, 29278654, 31932243, etc.). Hence, the originality of this manuscript needs to be clarified, and the differences between this study and other cohorts must be described and discussed to explain the relevant contribution to the field.

Response>

Thank you for your attention. We consider that the originality of this study is that this study is the first study revealed that within up-to-11 criteria is the only factor which contribute to CR of intermediate stage HCC beyond up-to-7 criteria treated by TACE. We added this information to discussion session.

-Previous reports have analyzed TACE treatment for intermediate HCC using different MTAs and immunotherapies. In this study, most cases of the CR group were treated with epirubicin (63) and cisplatin (3); however, there is not enough discussion of these results compared with studies that have used these drugs or others in TACE treatment for intermediate HCC. Therefore, the originality and relevance of the study must be clarified.

Response>

We added other information about outcomes of TACE using d combination therapy with systemic anticancer drug for intermediate stage HCC beyond up-to-7 criteria. There are few reports limited in beyond up-to-7 criteria. In addition, we added the originality of this study in discussion session.

-The study design has severe limitations, including the number of patients, single institution, etc. Therefore, require an exhaustive evaluation since the results only suggest its potential benefit in the studied cohort but not a clinical impact that it contributes to the treatment of intermediate HCC.

Response>

Thank you for your attention. We changed the title “Predictive factors of complete response to transarterial chemoembolization in intermediate stage hepato-cellular carcinoma beyond the up-to-7 criteria”. In addition we added your comments in limitation paragraph.

Minor comments

In line 157, I suggest describing the significate of PR (partial response) and SD (stable disease) abbreviations.

Response>

Thank you for your attention. We added the meanings of PR and SD.

Round 2

Reviewer 1 Report

I am unable to review this revised manuscript because the changes of authors' description was not shown in this resubmitted draft. All changes in content related to reviewer’s comments should be indicated in the letter of responses to reviewers and revised manuscript as blue/bold/underlined texts.

Reviewer 2 Report

The authors responded to each of my observations, so I consider the manuscript suitable for publication.

Round 3

Reviewer 1 Report

no more comment from me